# Potential Applications of Artificial Intelligence (AI) in Managing Polypharmacy in Saudi Arabia: A Narrative Review

**DOI:** 10.3390/healthcare12070788

**Published:** 2024-04-05

**Authors:** Safaa M. Alsanosi, Sandosh Padmanabhan

**Affiliations:** 1Department of Pharmacology and Toxicology, Faculty of Medicine, Umm Al Qura University, Makkah 24382, Saudi Arabia; 2BHF Glasgow Cardiovascular Research Centre, School of Cardiovascular and Metabolic Health, University of Glasgow, Glasgow G12 8QQ, UK; sandosh.padmanabhan@glasgow.ac.uk

**Keywords:** artificial intelligence, polypharmacy, Saudi Arabia, medication prescription

## Abstract

Prescribing medications is a fundamental practice in the management of illnesses that necessitates in-depth knowledge of clinical pharmacology. Polypharmacy, or the concurrent use of multiple medications by individuals with complex health conditions, poses significant challenges, including an increased risk of drug interactions and adverse reactions. The Saudi Vision 2030 prioritises enhancing healthcare quality and safety, including addressing polypharmacy. Artificial intelligence (AI) offers promising tools to optimise medication plans, predict adverse drug reactions and ensure drug safety. This review explores AI’s potential to revolutionise polypharmacy management in Saudi Arabia, highlighting practical applications, challenges and the path forward for the integration of AI solutions into healthcare practices.

## 1. Introduction

The practice of prescribing medications is crucial in improving health outcomes and requires extensive pharmacology knowledge and expertise. Polypharmacy, defined as the use of multiple medications by patients with complex health needs, is increasingly prevalent, especially in the context of the rising chronic disease rates in Saudi Arabia [1]. This presents significant challenges, including medication non-adherence and the potential for adverse drug reactions. With the Saudi Vision 2030 focusing on healthcare enhancement, artificial intelligence (AI) emerges as a critical tool in revolutionising medication management strategies [2,3].

The prescribing of medications is a practice used by doctors to improve symptoms and manage illnesses. It requires knowledge of prescribed drugs and the principles of clinical pharmacology and therapeutics [4]. Moreover, it calls for clinical expertise in evaluating the benefits and risks of medicines and in considering the evidence and factors related to a patient’s treatment. When dealing with diseases, a pharmacological approach is typically involved [5]. However, despite their effectiveness in controlling diseases, medications’ full advantages are often not realised, since 50% of patients do not adhere to their prescribed medication schedules [6].

Polypharmacy refers to the use of multiple medications by an individual who has complex health conditions and medical needs. The World Health Organization defines polypharmacy as ‘the administration of many drugs at the same time or the administration of an excessive number of drugs’ [7]. Most commonly, polypharmacy refers to five or more medications. However, there is no agreement in terms of definitions, and different definitions exist based on the length of treatment. For example, some definitions consider the use of two or more medications for over 240 days as ‘long-term use’, while others focus on the use of 5–9 medications for 90 days or more [8,9].

One systematic review examined the global prevalence of polypharmacy across different populations. It found that the prevalence of polypharmacy was 30.2% among community-dwelling individuals, 61.7% among hospitalised patients and 56.9% among institutionalised patients. These findings highlight the high occurrence of polypharmacy in healthcare settings [10]. A crucial aspect of the Saudi Vision 2030 plan is its focus on enhancing healthcare by fostering a strong culture centred on the safety and quality of medicines. In this regard, it is essential to foster a relationship between local and international drug manufacturers, as well as among Saudi regulatory authorities [11]. Moreover, the prevalence of chronic diseases such as diabetes, hypertension and cardiovascular ailments has increased in Saudi Arabia in recent years. Consequently, patients often require medications to manage their conditions [12].

The healthcare system in Saudi Arabia has undergone a significant transformation under Vision 2030, focusing on improving healthcare access, modernising facilities and encouraging private sector investment. Introducing e-health services and digital solutions like the SEHA Virtual Hospital has revolutionised healthcare delivery [3]. The Healthcare Sector Transformation Program has made the system more comprehensive, effective and integrated, prioritising innovation, financial sustainability and disease prevention. These advancements have improved public health and disease prevention and supported the goal of a long, healthy and productive life for all residents [13]. The government’s heavy investment in state-of-the-art hospitals and healthcare facilities, along with the integration of advanced medical technologies and digital solutions, has significantly enhanced the accessibility and quality of healthcare services, particularly in remote areas. These infrastructure improvements have been crucial in improving healthcare outcomes and ensuring better healthcare services for the Saudi population [11,14].

AI represents a broad field of computer science focused on creating smart machines capable of performing tasks that typically require human intelligence. AI encompasses various techniques and methodologies, with machine learning (ML) and deep learning being its most significant subsets. These technologies have found extensive application in healthcare, ranging from diagnostic procedures to treatment regimen planning and patient monitoring [15,16]. ML is a subset of AI that enables systems to learn from data, identify patterns and make decisions with minimal human intervention. It involves algorithms that improve the performance in a given task through increased exposure to data [17]. In the context of healthcare, particularly polypharmacy, ML can process vast amounts of patient data—clinical histories, genetic information and real-time health metrics—to identify risks, suggest personalised medication plans and predict potential adverse drug reactions [17,18].

Deep learning, a more complex subset of ML, uses neural networks with many layers to analyse large datasets. These models are particularly adept at recognising patterns in unstructured data, such as from text, images or sound [19]. Within healthcare, deep learning powers advanced diagnostic tools, including AI-assisted imaging analysis and NLP applications. Large language models (LLMs), such as Generative Pretrained Transformer (GPT), are a form of deep learning model that excels in understanding and generating human language [20,21]. Deep learning and LLMs can interpret medical literature, patient records and other textual data, aiding in the identification of relevant information for prescribing decisions and polypharmacy management. LLMs can summarise research findings, suggest medication alternatives based on the latest studies and even predict drug interactions by analysing extensive medical texts [19,22].

The healthcare industry recognises the importance of integrating artificial intelligence (AI)-driven tools into healthcare technology. AI has the potential to enhance various aspects of healthcare operations and delivery. One notable advantage is the potential cost savings that it can bring to the healthcare system [23]. Projections suggest that, by 2026, AI applications could reduce the annual healthcare costs in the United States by approximately USD 150 billion. These savings will primarily stem from a shift in the healthcare model, moving from a reactive approach focused on treating diseases to a proactive approach centred on health management. This transition will lead to fewer hospitalisations, fewer doctor visits and reduced treatment needs [24,25].

AI-based technologies will be crucial in promoting and maintaining a healthy lifestyle by enabling continuous monitoring and coaching. They will also facilitate earlier diagnosis, personalised treatments and more efficient follow-up care [23]. This will empower individuals to better manage their health and wellbeing. The market for AI in healthcare is expected to experience rapid growth, having reached an estimated value of USD 6.6 billion in 2021. This growth corresponds to a compound annual growth rate of 40% [24,26]. AI has the potential to revolutionise healthcare, not only in Saudi Arabia but globally. For example, it is a tool that healthcare professionals can use to improve patient outcomes in managing medical conditions [27]. In Saudi Arabia, as in other countries, healthcare professionals are dedicated to optimising medication regimens and reducing the unnecessary use of drugs [28]. AI can contribute to addressing the problem of polypharmacy by providing decision support tools to aid healthcare professionals in these efforts. Managing medications, monitoring patient progress and providing education are all demanding for healthcare professionals, and AI can help to reduce this burden [28,29].

To tackle the issue of polypharmacy, healthcare providers in Saudi Arabia employ strategies such as reconciling medications, conducting reviews of drugs, educating patients about their medications and enhancing communication among healthcare professionals [30]. However, healthcare practices and policies can change over time. Therefore, it is essential to start using AI in medication prescription practices, including managing medications, to reduce polypharmacy [31]. This review explores AI applications that can be used to manage polypharmacy in Saudi Arabia, considering the country’s focus on improving healthcare quality and safety, as outlined in the Saudi Vision 2030.

## 2. Methods and Materials

Published articles in the scientific literature examining AI’s effects on and potential applications in polypharmacy were identified and reviewed. We searched for relevant publications, including retrospective cohort studies, prospective studies, randomised controlled trials, systematic reviews and meta-analyses, using PubMed, Embase, Google Schooler and Web of Science. We used the following keywords and search terms: [Artificial Intelligence], [Machine Learning], [AI], [Polypharmacy], [Saudi Vision 2030] and [Saudi healthcare].

The results of the literature search relevant to the present topic were screened by title, keywords, abstract and then the whole publication. All searches were limited to the English language. Articles were not restricted by the date of publication.

## 3. Results 

### 3.1. Main Applications of AI in Polypharmacy

Physicians and researchers have studied polypharmacy extensively to reduce or prevent its negative consequences, such as drug events, drug interactions, patients’ non-adherence to medications and increased healthcare costs for both individuals and the healthcare system. In Saudi Arabia, as in other countries, polypharmacy can occur due to various factors, including comorbidities in an ageing population, the increased availability and accessibility of medicines, cultural beliefs, patient expectations and prescribing practices [32,33].

In addressing polypharmacy, AI offers a range of applications, from detecting drug–drug interactions to optimising medication adherence, as shown in Figure 1. By leveraging ML algorithms, healthcare professionals can identify potential adverse drug reactions, customise treatment plans and ensure more effective medication management. This section delves into specific AI applications, including real-world examples from Saudi Arabia, showcasing the technology’s potential to mitigate polypharmacy’s risks. In polypharmacy, AI and ML offer transformative potential in the following areas.

Drug–Drug Interaction Detection: AI algorithms can sift through extensive databases to identify potential adverse interactions between medications, reducing the risk of harmful side effects in patients with complex medication regimens.Personalised Medicine: By analysing genetic data, patient histories and current medications, ML models can help healthcare providers to tailor treatment plans to individuals, improving their efficacy and minimising unnecessary polypharmacy.Predictive Analytics: AI can forecast which patients are at risk of polypharmacy complications, allowing for pre-emptive adjustments to their treatment plans.Medication Adherence: AI-powered apps and devices can monitor patients’ adherence to medication schedules, providing reminders and alerts to both patients and healthcare providers to prevent the underuse or overuse of prescribed drugs.

### 3.2. Detection of Drug–Drug Interactions 

The issue of drug–drug interactions (DDIs) is a serious concern regarding healthcare and polypharmacy in Saudi Arabia. When patients take multiple medications, there is an increased risk of experiencing side effects, compromising patient safety [34]. The risks of polypharmacy can arise from the simultaneous use of anticoagulants, antihypertensives, antidiabetics and specific antibiotics. This situation becomes even more complicated when herbal or over-the-counter medications are added. The self-prescription of these medications in Saudi Arabia poses risks and challenges, including incorrect dosages, drug interactions and delayed medical care. The use of AI is crucial to ensuring safe and effective medication use. AI can play a vital role in educating the public about the appropriate use of these medications and their potential interactions and the importance of seeking professional advice for optimal health outcomes [35,36].

AI can be very helpful in this regard. AI algorithms can examine a patient’s list of medications and detect DDIs by considering known interactions among medications. AI systems can highlight polypharmacy issues and alert healthcare providers, enabling them to make informed decisions regarding medication management [37]. One study proposed a heterogeneous network-assisted inference framework (HNAI) for the prediction of DDIs. The HNAI models achieved an area under curve (AUC) of 0.67, as evaluated through fivefold cross-validation. The study focused on antipsychotic drugs and demonstrated that HNAI showed promise in uncovering DDIs during drug development and post-marketing surveillance [38].

ML algorithms can be trained to recognise interactions between medications. By analysing information regarding drug attributes, patient backgrounds and documented interactions, these algorithms can deliver polypharmacy alerts or suggestions to healthcare professionals, which can help to minimise the chances of adverse effects [39]. Healthcare professionals in Saudi Arabia already actively monitor medication regimens, conduct reviews and educate patients to minimise these risks. However, through research education initiatives and the integration of AI decision support tools, there is the potential to manage polypharmacy more effectively by improving the identification and management of DDIs [34,40].

### 3.3. Personalised Treatment Recommendations

Tailored treatments are gaining significance within Saudi Arabia’s healthcare system. Healthcare providers can improve treatment plans and enhance patient satisfaction by considering factors such as genetics, coexisting medical conditions, lifestyle choices and patient preferences [11]. AI has the capability to analyse patient data, such as their histories, laboratory results and medication profiles, to provide personalised treatment recommendations and avoid the potential negative effects of polypharmacy. By considering individual patient characteristics and potential drug interactions, AI systems can assist health professionals in optimising medication regimens and reducing the risks associated with polypharmacy [17,18]. For example, one systematic review examined 63 studies that utilised AI methods in precision cancer medicine. The review found that the included studies achieved high values for assessment indicators such as accuracy, sensitivity, specificity, precision, recall and area under the curve (AUC), with maximum values of 0.99, 1.00, 0.96, 0.98, 0.99 and 0.9929, respectively. The findings highlight the effective application of AI methods in personalised medicine, demonstrating their utility in improving cancer treatment and care [41].

It is possible to customise treatment plans for patients based on their traits, including genetics, demographics and medical backgrounds, by utilising ML to examine these factors alongside medication information [42]. The integration of health records and AI provides access to extensive patient information, thereby enabling the delivery of personalised treatment recommendations. Moreover, the implementation of AI in personalised medicine strategies has the potential to broadly enhance healthcare services, resulting in improved patient care and health outcomes throughout Saudi Arabia [43].

### 3.4. Prediction of Adverse Drug Reactions 

In Saudi Arabia, the regulatory authority responsible for ensuring drug safety is the Saudi Food and Drug Authority (SFDA), which has set up a system called pharmacovigilance to monitor and evaluate reports of adverse drug reactions (ADRs) [44]. By utilising datasets, AI can identify patterns and predict the likelihood of ADRs stemming from polypharmacy. Through the use of machine learning algorithms, AI systems can identify patients who are at risk of experiencing ill effects due to the simultaneous use of multiple medications [45]. For example, one study focused on estimating the likelihood of ADRs using various ML methods. The researchers built 14 predictive models using deep neural networks as part of their framework. These models achieved mean validation accuracy of 89.4%, indicating that the approach was successful in consistently predicting ADRs for a wide range of drugs [46]. This valuable information can assist healthcare providers in adjusting medication plans and closely monitoring patients’ wellbeing.

To enhance drug safety measures, the SFDA actively collaborates with organisations and databases that focus on pharmacovigilance. This partnership allows it to exchange information and identify risks associated with drug use [44]. ML algorithms can aid in recognising events linked to polypharmacy by examining data sources such as electronic health records and pharmacovigilance databases [40,47]. This approach can facilitate the identification and monitoring of medication-induced effects, enabling prompt intervention when necessary. AI has the potential to assist in pharmacovigilance by analysing real-world data, including media reports and posts about negative drug reactions. In doing so, it can help to identify safety issues related to polypharmacy [48,49]. This information can then be utilised in the drug development process, contributing to the development of safety profiles for medications used in polypharmacy.

The early detection of polypharmacy signals allows for investigation and regulatory actions or changes in prescribing practices. AI can automate various pharmacovigilance tasks, such as adverse event case processing, triaging and report generation [50,51]. Natural language processing algorithms can extract relevant polypharmacy information from adverse event reports, classify it according to the severity or causality and automate the generation of standardised reports. This automation can improve the efficiency, reduce the manual workload and enable faster response times [52].

### 3.5. Monitoring Medication Adherence

Ensuring that patients take their medications as prescribed is a crucial aspect of healthcare provision worldwide, including Saudi Arabia. Such adherence is crucial in managing health conditions and achieving the desired outcomes [2]. AI-powered technologies, such as automatic pill dispensers and medication reminder apps, can aid patients in following their medication schedules and send alerts if there is a risk of polypharmacy. These tools can send reminders to healthcare providers and patients to track their medication intake and provide feedback, thereby encouraging better adherence and reducing the chances of problems associated with polypharmacy [53].

For instance, one study developed a deep learning model for the binary classification of medication adherence using video monitoring in tuberculosis treatment. The selected automated deep learning models demonstrated moderate to high diagnostic properties and discriminative performance, with sensitivity ranging from 92.8% to 95.8%, specificity from 43.5% to 55.4%, F1 scores from 0.91 to 0.92, precision from 88% to 90.1% and an AUC from 0.78 to 0.85. The 3D ResNet model showed the highest precision, AUC and speed. The findings support the potential application of AI in predicting medication adherence through video frame classification [54].

ML algorithms have the capability of analysing data continuously, including signs, symptoms and how medications are being used, to provide real-time monitoring and feedback [55]. This is especially beneficial for patients with medication plans, as it allows for the detection of potentially problematic issues, such as not taking medications properly, adverse reactions to drugs and interactions between different drugs [56]. It is worth mentioning that the monitoring practices may differ depending on the healthcare setting and individual patient requirements. In Saudi Arabia, healthcare providers are embracing advancements to enhance medication adherence monitoring and patient outcomes.

### 3.6. Optimising Medication

Improving medication management in Saudi Arabia encompasses a range of strategies aimed at enhancing the safety, effectiveness, accessibility and outcomes of medications. It should be emphasised that achieving optimal medication therapy necessitates a well-coordinated approach involving healthcare professionals, policymakers, patients and other stakeholders [57]. AI has the potential to assist in optimising medication regimens by considering factors such as the effectiveness of drugs, patient safety and individual characteristics in managing polypharmacy [58]. By analysing data and clinical guidelines, AI algorithms can generate recommendations for the adjustment of medications (including the risk of polypharmacy). This may include optimising the dosages of drugs or minimising the number of medications [59].

Polypharmacy can lead to less-than-optimal medication plans, which can result in higher healthcare expenses, patients’ non-compliance with prescribed medications and potential harm [60]. Moreover, by identifying the possible interactions between drugs and opportunities to simplify drug use, ML algorithms can help healthcare providers to streamline medication plans and improve patient outcomes [61]. By implementing these strategies, Saudi Arabia has the potential to enhance the medication outcomes among its population, prioritise patient safety and ensure efficient healthcare delivery.

### 3.7. Real-Time Support for Decision Making

To improve drug decision making in Saudi Arabia, healthcare professionals can utilise clinical decision support systems. These computer-based tools offer evidence-based recommendations and information directly at the point of care. By implementing real-time drug decision support strategies, healthcare professionals can enhance medication safety, improve prescription practices and optimise patient outcomes [62]. In addition, AI can provide healthcare professionals with real-time decision support during patient care. By integrating electronic health records (EHRs) and leveraging patient data, AI systems can flag potential issues related to polypharmacy, such as identifying prescriptions, contraindications and inappropriate dosages. Such support can enable clinicians to make decisions about medication management during consultations with patients [27,63].

Furthermore, ML algorithms have the capability to sort patients into polypharmacy risk categories based on their medication histories and medical records, as well as other pertinent factors [64]. This sorting can assist healthcare professionals in identifying patients who are likelier to experience medication-related complications or who would benefit from monitoring and medication management interventions [64,65]. However, there are some limitations in the real-time support provided for drug decision making using AI in healthcare, including a lack of interpretability, insufficient data quality and availability, limited generalisability, ethical and legal considerations, integration challenges, a reliance on historical data and the need for human–AI collaboration. Addressing these limitations requires ongoing research, collaboration between AI experts and healthcare professionals, validation methodologies and a comprehensive understanding of the ethical and legal implications [66].

AI systems greatly enhance the knowledge and decision-making abilities of prescribers, which ultimately leads to better medication management. By incorporating ML into decision support systems, healthcare professionals in Saudi Arabia can receive real-time recommendations and guidelines to manage polypharmacy. These systems can alert providers about drug interactions, suggest medications and propose dosage adjustments based on individual patient factors [26]. For example, a data mart was developed for the management of COVID-19 patients, based on a cohort study of 5528 patients. The dashboard supports management at the hospital, ward and individual care levels. The data mart allows for predictive modelling, identifying clinical phenotypes and fostering hospital networks. Real-time updated dashboards from the data mart can enhance the understanding of COVID-19’s epidemiology and clinical features, aiding in future wave predictions and pandemic management [67].

The implementation considerations once an ML model is validated and ready for clinical use include the following.

Interoperability: The system must be compatible with the existing EHR and healthcare IT infrastructure.User Interface: Alerts and recommendations should be presented in a user-friendly manner that integrates seamlessly into healthcare professionals’ workflows.Privacy and Security: Patient data used in the training and operation of the model must be handled according to strict privacy and security standards.Regulatory Compliance: The development and deployment of the system must comply with the relevant healthcare regulations and standards.

Through the integration of data sources, such as health records, genomics and patient-reported outcomes, ML algorithms offer decision support. By analysing this wealth of information, AI algorithms can provide medication recommendations that consider factors such as effectiveness, safety and patient preferences. This comprehensive approach aids healthcare providers in making informed decisions, especially when it comes to managing medications in cases of polypharmacy [68,69].

### 3.8. Predictive Analysis

The predictive analysis of drugs in Saudi Arabia involves using data analytics and ML methods to forecast drug-related aspects, including drug effectiveness, adverse reactions, medication adherence and patterns of drug utilisation. Through the analysis of datasets, AI algorithms can identify patterns and predict risks that are specific to individual patients who are taking multiple medications simultaneously [70]. By examining patients’ demographics, medical histories, genetic information and current medication profiles, AI can identify individuals who might be more susceptible to polypharmacy. By utilising ML algorithms, we can examine sets of health data, such as health records, clinical trials and the medical literature, enabling us to identify patterns and make predictions regarding the probability of drug interactions or side effects [69].

The knowledge gained from predictive analysis empowers healthcare providers to make informed decisions when prescribing medications. In addition, ML can be utilised to predict the likelihood of events or treatment outcomes associated with medications [71]. One quality improvement project utilised ML techniques to predict and reduce outpatient MRI appointment no-shows. Developed with XGBoost, the predictive model achieved an ROC AUC of 0.746 and an optimised F1 score of 0.708. Implementing telephone call reminders for patients identified as having the highest risk of no-shows reduced the overall no-show rate from 19.3% to 15.9%, demonstrating a 17.2% improvement. ML predictive analytics can be integrated into routine healthcare workflows to enhance healthcare delivery and address complex problems involving human behaviour [72].

By examining data and outcomes through ML models, we can identify the factors that contribute to successful or unsuccessful treatment. This valuable information aids healthcare providers in making decisions about medications, dosages and for how long therapies should be administered [73]. However, it is crucial to emphasise that successful drug predictive analysis relies on access to high-quality, comprehensive data; ethical practices regarding privacy; and the validation of models using real-world data. By utilising these analytical techniques, Saudi Arabia can enhance medication safety, improve treatment outcomes and optimise the allocation of healthcare resources in the realm of drug therapy.

### 3.9. Remote Education and Telemedicine

In Saudi Arabia, there has been a rise in the significance of education and telemedicine, primarily driven by the COVID-19 pandemic. These technological advancements have improved the accessibility of education and healthcare services in underserved or isolated areas [74]. For example, AI-powered telemedicine platforms provide online sessions and consultations related to polypharmacy with healthcare professionals. Patients can participate in classes, webinars and support groups to gain polypharmacy knowledge related to their conditions, treatments and self-care strategies [75]. In addition, AI technology can enable personalised content delivery, interactive sessions and patient engagement in distance learning initiatives. Importantly, AI-driven tools can empower patients by involving them in polypharmacy management processes [76].

For instance, virtual assistants or chatbots equipped with AI capabilities can offer information about polypharmacy to address patient queries and even send them reminders to take their medicines. These technologies have the potential to assist patients in understanding their medication routines, managing side effects and emphasising the importance of adhering to medications, thereby contributing to reducing the risks associated with polypharmacy [53,61]. Online education and telemedicine in Saudi Arabia have immense potential in expanding access to education and healthcare, thereby enhancing patient outcomes and optimising resource allocation. This is especially valuable during times of crisis, such as pandemics, or in regions where the healthcare infrastructure is limited [77].

One study conducted a bibliometric analysis of AI in telemedicine, examining publication trends, countries/regions, authors, journals, influential articles and keyword usage. The findings revealed an annual increase of approximately 42.1% in articles published on AI in telemedicine. The United States and China were the leading contributors, and keywords such as ‘machine learning’ and ‘digital health’ were prominent. The study identified mobile health as a promising area for future research and provided insights for the advancement of AI in telemedicine [78]. However, there are several potential limitations of remote education and telemedicine, including limited access to technology, technical issues and connectivity problems, a lack of personal interaction and hands-on experience, privacy and security concerns, limited non-verbal cues, communication challenges, skill and technological literacy requirements, inadequate infrastructure and limited resources. These limitations can create disparities in access, disrupt sessions, hinder effective communication and raise privacy concerns, and they can be addressed through policy, infrastructure development and digital literacy programs [74,79].

### 3.10. Designing Clinical Trials

The use of clinical trials to address polypharmacy in Saudi Arabia requires compliance with regulations to protect patient safety and uphold ethical standards. It is crucial to include different populations to improve the applicability of study findings. Moreover, establishing partnerships with healthcare institutions and research centres provides the needed resources and expertise [26,80]. However, it is important to mention that while AI can provide insights and assistance in drug development and medication management, it should always be used alongside human expertise and rigorous scientific validation. The incorporation of AI into the drug development process necessitates validation and regulatory supervision to ensure the safety of patients and to achieve the best healthcare outcomes [37].

By designing clinical trials and selecting patients, AI has the potential to assist in assessing the safety and effectiveness of the drugs used in polypharmacy. By analysing data and the medical literature, AI algorithms can identify groups of patients who may benefit greatly from certain combinations of drugs. This can facilitate efficient clinical trials, ultimately resulting in faster drug development processes and regulatory approval [64]. By following regulations prioritising diversity in recruitment efforts and fostering collaboration, Saudi Arabia can advance knowledge and enhance healthcare outcomes through thoughtfully designed clinical trials.

## 4. Discussion 

Although AI shows considerable promise in addressing polypharmacy concerns in Saudi Arabia and worldwide, there are challenges that need to be addressed. These challenges include ensuring access to high-quality data, considering regulations related to AI usage in healthcare settings, maintaining transparency regarding algorithms and guaranteeing access to healthcare technologies driven by AI. Collaboration between healthcare professionals, AI experts and researchers from fields specialising in AI is essential to harness AI’s potential when it comes to effectively managing polypharmacy. For instance, one systematic review of the potential of algorithms and the adoption of AI in enhancing medication management in primary care revealed that studies present varying perspectives on the successful reduction of medication errors. About 71% of the studies in the review documented a decrease in medication errors. These findings provide further support for the hypothesis that AI serves as a significant tool in ensuring patient safety [81].

AI, including ML, can help healthcare professionals to deal with the challenges of polypharmacy by offering insights and decision support. Several studies have demonstrated that AI-based interventions can improve medication adherence, reduce adverse drug events and enhance medication safety. AI algorithms have been used to identify DDIs, predict medication responses and personalise treatment plans, leading to better therapeutic outcomes. However, specific statistical data on patient outcomes resulting from AI interventions in polypharmacy may vary depending on the study design and context [26,53]. For instance, in a study using an AI-aided rational drug use web assistant, 88.9% of nursing home patients experienced polypharmacy, with an average of 6.96 ± 2.94 drugs per patient. Risky drug interactions were found in 89.9% of the patients, contraindicated interactions in 20.2% and potentially inappropriate drug use in 86.9%. Discontinuing 83 medications resulted in estimated cost savings of 9.1% per month. The rational drug use web assistant, integrated with AI, significantly reduced the number of drugs and instances of polypharmacy. The study concluded that this cost-effective application can benefit family physicians and their geriatric patients [82].

However, it is important to remember that healthcare professionals are still responsible for managing polypharmacy. AI algorithms should be seen as tools that enhance judgment, rather than as a replacement for human expertise. It is crucial to validate the data and continuously assess the algorithms. In addition, to ensure the successful implementation of ML in healthcare, privacy concerns and ethical considerations must be addressed [61,66]. Integrating AI technologies requires the careful consideration of data privacy and security and ethical concerns to maintain confidentiality and trust. The adoption of AI in the Saudi Arabian healthcare system will likely progress gradually through the influence of advancements, research findings and healthcare policies. Effectively utilising AI while prioritising safety and wellbeing requires ongoing collaboration between healthcare providers, policymakers and researchers [83].

AI is transforming the healthcare landscape in Saudi Arabia, bringing advancements in the diagnosis of illnesses, improving the effectiveness of treatments and enhancing medical outcomes. Through the integration of AI-powered medical imaging analysis and predictive algorithms, disease detection and risk assessment have become faster and more precise. The healthcare sector in Saudi Arabia is actively adopting AI to optimise the allocation of resources, tailor treatments to patients’ needs and improve overall healthcare services.

As healthcare continues to evolve, AI technologies are poised to become an integral component of patient care strategies. The predictive capabilities of ML models offer a proactive approach to identifying potential adverse drug interactions before they occur, enabling healthcare providers to design medication regimens that are safe and effective for each individual patient [24]. While AI has the potential to optimise medication use and prevent adverse events, its precise cost-effectiveness in polypharmacy has not been documented extensively. AI interventions that improve adherence, prevent hospitalisation or reduce unnecessary medication use can potentially lead to cost savings in healthcare systems. However, comprehensive economic evaluations and long-term cost-effectiveness studies are needed to provide more precise statistics in this regard [84,85].

As shown in Figure 2, for healthcare professionals and policymakers, understanding the basics of AI and ML is crucial in leveraging these technologies to address the challenges of polypharmacy. By integrating AI tools into healthcare systems, professionals can enhance medication safety, improve patient outcomes and streamline the prescribing process [84]. However, successful implementation requires continuous education in these technologies, ethical considerations and collaborative efforts between AI experts, healthcare providers and policymakers to ensure that these tools are used effectively and responsibly.

Looking ahead, the continued advancement and integration of AI in healthcare will likely focus on several key areas.

Enhanced Personalisation: AI will drive the shift towards more personalised medicine, where treatments and medication regimens are tailored to the individual’s genetic makeup, lifestyle and specific health conditions.Interoperability and Integration: The seamless integration of AI tools with existing EHRs and healthcare systems will be crucial. This will ensure that AI-driven insights are readily accessible to healthcare providers, facilitating informed decision making.Ethical AI Use: As AI takes on a more prominent role in healthcare, ethical considerations, including patient privacy, data security and algorithmic transparency, will become increasingly important. It will be essential to establish guidelines and standards for the ethical use of AI to maintain trust and protect patient rights.Education and Collaboration: Educating healthcare professionals about AI and its potential applications in polypharmacy management will be key to AI’s successful implementation. Furthermore, fostering collaboration between AI researchers, healthcare providers and patients will ensure that the AI solutions are effectively tailored to meet the needs of those whom they aim to serve.Continuous Innovation and Research: Ongoing research and innovation will be vital in expanding the capabilities of AI in healthcare. This includes developing new algorithms, refining existing models and exploring novel applications to address the complexities of polypharmacy and medication adherence.

## 5. Conclusions

The future of AI in healthcare is bright, particularly in the context of polypharmacy and medication adherence. With its potential to transform patient care, AI offers a path towards safer, more effective and personalised treatment strategies. However, realising this potential will require concerted efforts across the healthcare ecosystem to address challenges related to integration, ethics, education and ongoing innovation. As we move forward, it is clear that AI will play a pivotal role in shaping the future of healthcare, promising improved outcomes for patients worldwide.

## Figures and Tables

**Figure 1 healthcare-12-00788-f001:**
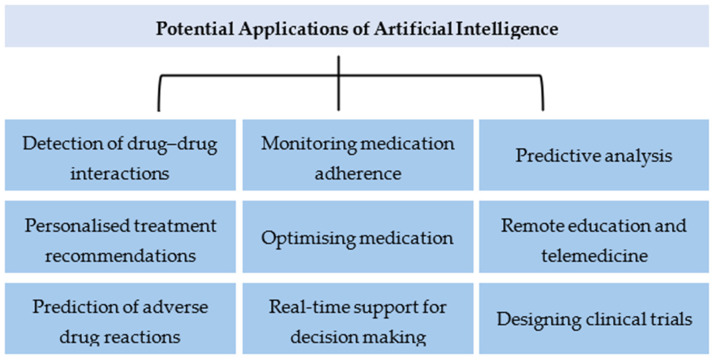
Potential applications of AI in managing polypharmacy in Saudi Arabia.

**Figure 2 healthcare-12-00788-f002:**
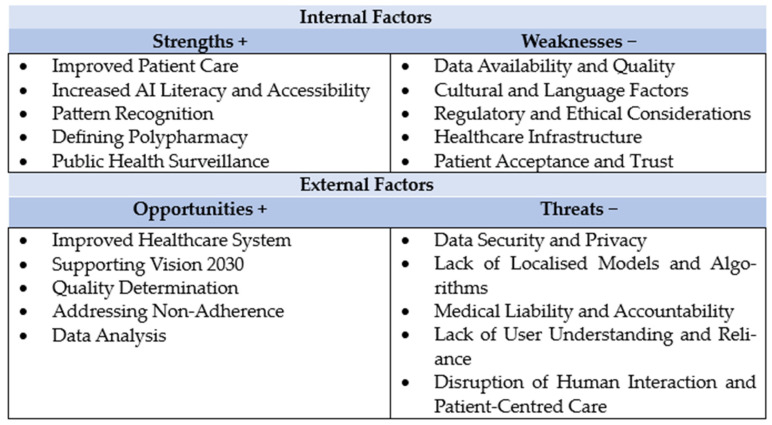
SWOT analysis of using AI in managing polypharmacy in Saudi Arabia.

## Data Availability

Not applicable.

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
