# Peer review of "Potential Applications of Artificial Intelligence (AI) in Managing Polypharmacy in Saudi Arabia: A Narrative Review"

_healthcare, 2024, doi:10.3390/healthcare12070788_

Round 1

Reviewer 1 Report

Comments and Suggestions for Authors

Dear authors,

I would like to express my gratitude to the journal for giving me the opportunity to review the manuscript entitled "Potential Applications of Artificial Intelligence (AI) in Managing Polypharmacy in Saudi Arabia." The article is a review aiming to explore the potential applications of artificial intelligence in Saudi Arabia concerning the management of polypharmacy. The topic is of significant interest as it represents a hotly debated issue with substantial implications for public health. Therefore, I commend the authors for recognizing its importance.

However, there are several critical points in the manuscript that need to be addressed before it can be considered for publication:

Introduction:

1. It is imperative to provide a clearer definition of the concept of polypharmacy. How many medications must a person take for the term "polypharmacy" to be applicable?

2. A more comprehensive consideration of the international literature is necessary. I recommend expanding the number of citations in the background section since this topic has been analyzed in various countries.

3. The objective of the study needs to be better elucidated. The current statement, "This review explores AI applications that can be used to manage polypharmacy in Saudi Arabia," is too concise and does not fully convey the study's purpose.

Materials and Methods:

1. This section is missing from the study, making it difficult to understand the review process. I suggest analyzing and adapting a flowchart similar to the one used in the study on polypharmacy (10.1136/bmjopen-2021-057399).

2. Specify the type of review being conducted. This will help readers better understand the methodological approach of the study.

Results:

1. Everything written after this point should be incorporated into the Results section.

Discussion:

1. The analysis of the obtained results and their comparison with international literature (references: 10.15537/smj.2019.2.23933, 10.1136/bmjopen-2022-065301, and 10.3390/healthcare9080961) is missing.

I hope you will consider the necessary suggestions so that the article can be taken into consideration for publication, as the topic is highly compelling.

Comments on the Quality of English Language

I suggest a revision of your manuscript to enhance its clarity.

Author Response

Reviewer 1  

 Thank you – We really appreciate your valuable feedback and comments.

Introduction:

  1. It is imperative to provide a clearer definition of the concept of polypharmacy. How many medications must a person take for the term "polypharmacy" to be applicable?

  • Response: Done -Polypharmacy refers to the use of multiple medications by an individual who has complex health conditions and medical needs. The World Health Organization defines polypharmacy as ‘the administration of many drugs at the same time or the administration of an excessive number of drugs’ (7). Most commonly, polypharmacy refers to five or more medications. However, there is no agreement in terms of definitions, and different definitions exist based on the length of treatment. For example, some definitions consider the use of two or more medications for over 240 days as ‘long-term use’, while others focus on the use of 5–9 medications for 90 days or more (8, 9).

  1. A more comprehensive consideration of the international literature is necessary. I recommend expanding the number of citations in the background section since this topic has been analyzed in various countries.
  • Response: Done

  1. The objective of the study needs to be better elucidated. The current statement, "This review explores AI applications that can be used to manage polypharmacy in Saudi Arabia," is too concise and does not fully convey the study's purpose.

  • Response: Done - This review explores AI applications that can be used to manage polypharmacy in Saudi Arabia, considering the country’s focus on improving healthcare quality and safety, as outlined in Saudi Vision 2030.

Materials and Methods:

  1. This section is missing from the study, making it difficult to understand the review process. I suggest analyzing and adapting a flowchart similar to the one used in the study on polypharmacy (10.1136/bmjopen-2021-057399).

  • Response: Done - Methods and materials : Published articles in the scientific literature examining AI’s effects on and potential applications in polypharmacy were identified and reviewed. We searched for relevant publications, including retrospective cohort studies, prospective studies, randomised controlled trials, systematic reviews and meta-analyses using PubMed, Embase, Google Schooler and Web of Science. We used the following keywords and search terms: [Artificial Intelligence], [Machine Learning], [AI], [Polypharmacy], [Saudi Vision 2030] and [Saudi healthcare]. The results of the literature search relevant to the present topic were screened by title, keywords, abstract and then the whole publication. All searches were limited to the English language. Articles were not restricted by the date of publication.

  1. Specify the type of review being conducted. This will help readers better understand the methodological approach of the study.
  • Response : Done - Narrative review

Results:

  1. Everything written after this point should be incorporated into the Results section.
  • Response : Done

Discussion:

  1.   The analysis of the obtained results and their comparison with international literature (references: 10.15537/smj.2019.2.23933, 10.1136/bmjopen-2022-065301, and 10.3390/healthcare9080961) is missing.
  • Response – Done: Collaboration between healthcare professionals, AI experts and researchers from fields specialising in AI is essential to harness AI’s potential when it comes to effectively man-aging polypharmacy. For instance, one systematic review of the potential of algorithms and the adoption of AI in enhancing medication management in primary care revealed that studies have presented varying perspectives on the successful reduction of medica-tion errors. About 71% of the studies in the review documented a decrease in medication errors. These findings provide further support for the hypothesis that AI serves as a sig-nificant tool in ensuring patient safety (79).

I hope you will consider the necessary suggestions so that the article can be taken into consideration for publication, as the topic is highly compelling.Comments on the Quality of English Language ; I suggest a revision of your manuscript to enhance its clarity.

  • Response: Done – the review sent for proofreading (attached certificate)

Reviewer 2 Report

Comments and Suggestions for Authors

Potential Applications of Artificial Intelligence (AI) in Managing Polypharmacy in Saudi Arabia

The authors highlight potential use of AI/ML in addressing medical issues related to multiple drug interactions in patients in Saudi Arabia (SA). The manuscript outlines several points, but lacks quantitative details of current medical infrastructure making it difficult to access how the proposed claims would materialize.

Comments:

Line 26: Medication non-adherence is a patient driven issue that can originated due to several reasons. There are a lot of research related to this, but not sure how it is associated with AI/ML. If authors want to proceed in this direction, they need to provide additional details. Similarly provide more details on “potential for adverse drug reactions” with references.

Line 35: “ …realised because 50% of patients do not adhere to their prescribed medication schedule (3).” This statement is misleading. The reference given is very specific to certain facility, age group in a specific country and can’t be generalized.

Line 88: language. They can interpret medical literature, patient records, and other textual data, aiding in the identification of relevant information for prescribing decisions and polypharmacy management. LLMs can summarize research findings, suggest medication alternatives based on the latest studies, and even predict drug interactions by analyzing extensive medical texts.

While LLM are greatly capable of many things, please provide references to your claim.

Main Applications of AI in Polypharmacy: In this segment, statistical data and accurate examples are missing.

Line 128: please provide details on how hearbal and OTC medications complicate? how AI will be useful in this regard?

It is not clear how AI can work specifically in OTC and herbal medication that do not come from prescription.

These medications are taken at random without any control. How these information will reach to the AI to alert anyone?

the lack of proper statistical data make this paragraph vague. It is not clear what are the current incident rates and what the effectiveness of the currently practiced measure.

Line 150: Please back the claims with appropriate statistics.

Line 122: while the author's claims are all valid. The lack of quantitative knowledge of the current infrastructure makes it difficult to assess how the proposed claim of AI would work.

Overall, the manuscript lacks several citations to the claims and statistical details.

Comments on the Quality of English Language

Several repeat statements and language similar to machine-generated (hallucinating language).

Author Response

Reviewer 2

Thank you – We really appreciate your valuable feedback and comments.

The authors highlight potential use of AI/ML in addressing medical issues related to multiple drug interactions in patients in Saudi Arabia (SA). The manuscript outlines several points, but lacks quantitative details of current medical infrastructure making it difficult to access how the proposed claims would materialize.

  • Line 26: Medication non-adherence is a patient driven issue that can originated due to several reasons. There are a lot of research related to this, but not sure how it is associated with AI/ML. If authors want to proceed in this direction, they need to provide additional details. Similarly provide more details on “potential for adverse drug reactions” with references.
  • Response: Done -  Added to monitoring medication adherence and prediction of adverse drug reactions sections

  • Line 35: “ …realised because 50% of patients do not adhere to their prescribed medication schedule (3).” This statement is misleading. The reference given is very specific to certain facility, age group in a specific country and can’t be generalized.
  • Response _ Done : The reference were corrected [ Brown MT, Bussell JK. Medication adherence: WHO cares? Mayo Clin Proc. 2011 Apr;86(4):304-14. doi: 10.4065/mcp.2010.0575. Epub 2011 Mar 9. PMID: 21389250; PMCID: PMC3068890]

  • Line 88: language. They can interpret medical literature, patient records, and other textual data, aiding in the identification of relevant information for prescribing decisions and polypharmacy management. LLMs can summarize research findings, suggest medication alternatives based on the latest studies, and even predict drug interactions by analyzing extensive medical texts.
  • Response: Done – Rephrased ( the review sent for proofreading  (attached certificate)

  • While LLM are greatly capable of many things, please provide references to your claim.
  • Response: Done – we added the references

  • Main Applications of AI in Polypharmacy: In this segment, statistical data and accurate examples are missing.
  • Response: Done - statistical data were added ( written in blue )

  • Line 128: please provide details on how hearbal and OTC medications complicate? how AI will be useful in this regard? It is not clear how AI can work specifically in OTC and herbal medication that do not come from prescription. These medications are taken at random without any control. How these information will reach to the AI to alert anyone?
  • Response: Done- The risks of polypharmacy can arise from the simultaneous use of anticoagulants, anti-hypertensives, antidiabetics and specific antibiotics. This situation becomes even more complicated when herbal or over-the-counter medications are added. Self-prescription of these medications in Saudi Arabia poses risks and challenges, including incorrect dosage, drug interactions and delayed medical care. Using AI is crucial to ensuring safe and effec-tive medication use. AI can play a vital role in educating the public about the appropriate use of these medications, potential interactions and importance of seeking professional advice for optimal health outcomes (33, 34).

  • Line 150: Please back the claims with appropriate statistics.
  • Response: Done- written in blue

  • Line 122: while the author's claims are all valid. The lack of quantitative knowledge of the current infrastructure makes it difficult to assess how the proposed claim of AI would work.
  • Response: Done- written in blue

Overall, the manuscript lacks several citations to the claims and statistical details.

  • Response: Done- written in blue

Reviewer 3 Report

Comments and Suggestions for Authors

The review article by Alsanosi and Padmanabhan addresses a very interesting topic, which is the contribution that the application of artificial intelligence (AI) can make in polipharmacy to prevent drug interferences as well as adverse effects and to optimize the medication plan. However, in my opinion, the review article remains too theoretical and superficial, not going deep into the AI aspects and not giving details and examples on the methodologies and results of AI applications, neither in general nor in the specific context of Saudi Arabia, as expected from the title. In many cases, the article cites other review articles, some of them much more precise and detailed, instead of reporting specific studies, discussing their limitations, methods and achievements.

Furthermore, the articles completely lack tables and figures, which are essential to focus and clarify the main points of the review.

Comments on the Quality of English Language

I do not have any specific comment.

Author Response

Reviewer 3

Thank you – We really appreciate your valuable feedback and comments.

The review article by Alsanosi and Padmanabhan addresses a very interesting topic, which is the contribution that the application of artificial intelligence (AI) can make in polipharmacy to prevent drug interferences as well as adverse effects and to optimize the medication plan.

  • However, in my opinion, the review article remains too theoretical and superficial, not going deep into the AI aspects and not giving details and examples on the methodologies and results of AI applications, neither in general nor in the specific context of Saudi Arabia, as expected from the title. In many cases, the article cites other review articles, some of them much more precise and detailed, instead of reporting specific studies, discussing their limitations, methods and achievements.
  • Response: Done – written in blue

  • Furthermore, the articles completely lack tables and figures, which are essential to focus and clarify the main points of the review.
  • Response: Done –Added to the manuscript

Reviewer 4 Report

Comments and Suggestions for Authors

The paper discusses the potential of AI use for polypharmacy management in different subfileds. Although the structure is well-organized and the logistic is clear, the absence of quantitative analysis makes the arguments less convincing. Without those analysis on output metrics, i.e., patient outcomes and cost effctiveness, the review's conclusion is not persuasive. 

Author Response

Reviewer 4

Thank you – We really appreciate your valuable feedback and comments.

  1. The paper discusses the potential of AI use for polypharmacy management in different subfileds. Although the structure is well-organized and the logistic is clear, the absence of quantitative analysis makes the arguments less convincing. Without those analysis on output metrics, i.e., patient outcomes and cost effctiveness, the review's conclusion is not persuasive. 
  • Response: Done – Written in blue (discussion part )

Reviewer 5 Report

Comments and Suggestions for Authors

Hello.

Thank you for your submission. I feel that your manuscript was a great summary.  But I think it could be better if you included more examples for each section with data. Currently, everything is generalized.  For example, line 133, you mention that ML can recognize DDIs, but can it differentiate from clinically significant interactions or just general interaction. It would help the reader if you provides an example of an interaction that ML can identify. 

The other aspect is that each of the section really paints a rosy picture of AI without discussion on potential draw backs. Identifying drug interactions is important but there is a potential for too many alerts when the process first starts.  Also what about intentional drug interactions such as darunavir and ritonavir for boosted protease inhibitor therapy. These potential alerts can actually slow down the workflow process if the provider has to check and address each issue. 

For the section on prediction of adverse effects. The comments are also general. It would be ideal if the paper had examples of how accurate are we with regards ML and identifying ADRs. Are the models as accurate as traditional processes?

There was also a discussion of improving efficiency and faster response times. It would be nice to know how fast is the process. The flip side of the efficiency is outcomes. Does increasing the reporting speed really impact outcomes? 

under medication optimization. please provide examples.

Under real time support for decision making. please include some limitations that is not overcome by AI such as person power to act on the result/alert.

remote education and telemedicine. include potential limitations. Providing information does not equal uptake from the patients.

Overall, great review.

Author Response

Reviewer 5

Thank you – We really appreciate your valuable feedback and comments.

Thank you for your submission. I feel that your manuscript was a great summary.  But I think it could be better if you included more examples for each section with data. Currently, everything is generalized.

  1. For example, line 133, you mention that ML can recognize DDIs, but can it differentiate from clinically significant interactions or just general interaction. It would help the reader if you provides an example of an interaction that ML can identify. 
  • Response: Done –Added to Detection of Drug-Drug Interactions part
  1. The other aspect is that each of the section really paints a rosy picture of AI without discussion on potential draw backs. Identifying drug interactions is important but there is a potential for too many alerts when the process first starts.  Also what about intentional drug interactions such as darunavir and ritonavir for boosted protease inhibitor therapy. These potential alerts can actually slow down the workflow process if the provider has to check and address each issue. 
  • Response: Done – Drawbacks and challenges were added to the discussion part
  1. For the section on prediction of adverse effects. The comments are also general. It would be ideal if the paper had examples of how accurate are we with regards ML and identifying ADRs. Are the models as accurate as traditional processes?
  • Response: Done –Added to Prediction of Adverse Drug Reactions part
  1. There was also a discussion of improving efficiency and faster response times. It would be nice to know how fast is the process. The flip side of the efficiency is outcomes. Does increasing the reporting speed really impact outcomes? 
  • Response: Done – The early detection of polypharmacy signals allows for investigation and regulatory actions or changes in prescribing practices. AI can automate various pharmacovigilance tasks, such as adverse event case processing, triaging and report generation (48, 49). Nat-ural language processing algorithms can extract relevant polypharmacy information from adverse event reports, classify it according to severity or causality and automate the gen-eration of standardised reports. This automation can improve efficiency, reduce manual workload and enable faster response times (50).
  1. under medication optimization. please provide examples.
  • Response: Done - Added to medication optimization part
  1. Under real time support for decision making. please include some limitations that is not overcome by AI such as person power to act on the result/alert.
  • Response: Done- However, there are some limitations in real-time support for drug decision making using AI in health care, including a lack of interpretability, insufficient data quality and availability, limited generalisability, ethical and legal considerations, integration challenges, re-liance on historical data and the need for human–AI collaboration. Addressing these limitations requires ongoing research, collaboration between AI experts and healthcare professionals, validation methodologies and a comprehensive understanding of ethical and legal implications(64).
  1. remote education and telemedicine. include potential limitations. Providing information does not equal uptake from the patients. Overall, great review.
  • Response: Done- However, there are several potential limitations of remote education and telemedicine, including limited access to technology, technical issues and connectivity problems, a lack of personal interaction and hands-on experience, privacy and security concerns, limited non-verbal cues, communication challenges, skill and technological literacy requirements, inadequate infrastructure and limited resources. These limitations can create disparities in access, disrupt sessions, hinder effective communication and raise privacy concerns, and they can be addressed through policy, infrastructure development and digital literacy programs (72, 77).

Round 2

Reviewer 1 Report

Comments and Suggestions for Authors

Dear authors, I have carefully read the second version of your manuscript. Congratulations, it has improved in all its parts; therefore, in my opinion, the article is now suitable for publication.

Author Response

We want to express our gratitude for your valuable feedback. Your comments in the first round were greatly appreciated and significantly enhanced the manuscript.

Reviewer 2 Report

Comments and Suggestions for Authors

The authors have addressed most of the review comments.

The concern remains around the applicability of this manuscript in the context of Saudi Arabia (SA). No details about the healthcare condition and current infrastructure in SA are provided. 

Many of the proposals in the manuscript seem to be more of an informatics/health informatics solution (HIT) than purely AI.

It might be more beneficial if the authors turn the manuscript in the direction of HIT and keep AI as a significant contributor once appropriate data are available.

Author Response

Thank you – We really appreciate your valuable feedback and comments.

  • The concern remains around the applicability of this manuscript in the context of Saudi Arabia (SA). No details about the healthcare condition and current infrastructure in SA are provided. 

Response - Done – We added:

The healthcare system in Saudi Arabia has undergone a significant transformation under Vision 2030, focusing on improving healthcare access, modernizing facilities, and encouraging private sector investment. Introducing e-health services and digital solutions like the SEHA Virtual Hospital has revolutionized healthcare delivery(3). The Healthcare Sector Transformation Program has made the system more comprehensive, effective, and integrated, prioritizing innovation, financial sustainability, and disease prevention. These advancements have improved public health and disease prevention and supported the goal of a long, healthy, and productive life for all residents(13). The government's heavy investment in state-of-the-art hospitals and healthcare facilities, along with the integration of advanced medical technologies and digital solutions, has significantly enhanced the accessibility and quality of healthcare services, particularly in remote areas. These infrastructure improvements have been crucial in improving healthcare outcomes and ensuring better healthcare services for the Saudi population(11, 14).

  • Many of the proposals in the manuscript seem to be more of an informatics/health informatics solution (HIT) than purely AI.It might be more beneficial if the authors turn the manuscript in the direction of HIT and keep AI as a significant contributor once appropriate data are available.

Response – We appreciate the valuable feedback you gave us. Your comments during the first round significantly improved the quality of the manuscript. However, we hold different perspectives on this matter.

Reviewer 3 Report

Comments and Suggestions for Authors

I appreciate the effort the authors put into revising their article. The review is much improved and contains more data and scientific detail on the application of AI than the previous version. The authors have also mentioned the strategy used to select the articles to be discussed in the review. The discussion is also improved and describes the current challenges and perspectives of AI application in polypharmacy.

However, in my opinion, at least one more figure or table could be added, besides Figure 1, to improve the readability and clarity of the message. My suggestion would be to add a figure representing a SWOT analysis that  summarizes the strengths, weaknesses, opportunities, and threats associated with AI application in this context.

Minor points:

- Subsections 1.1. and 1.2. are very short and both refer to AI. For this reason, I would suggest eliminating the division, combining both parts, and placing them in the Introduction where AI is first introduced (e.g. lines 56-57).

-Line 186: Perhaps this is "the area under the receiver operating characteristic (ROC) curve, that is AUC". Please check.

Author Response

We appreciate the valuable feedback you gave us. Your comments during the first round significantly improved the quality of the manuscript.

I appreciate the effort the authors put into revising their article. The review is much improved and contains more data and scientific detail on the application of AI than the previous version. The authors have also mentioned the strategy used to select the articles to be discussed in the review. The discussion is also improved and describes the current challenges and perspectives of AI application in polypharmacy.

  • However, in my opinion, at least one more figure or table could be added, besides Figure 1, to improve the readability and clarity of the message. My suggestion would be to add a figure representing a SWOT analysis that  summarizes the strengths, weaknesses, opportunities, and threats associated with AI application in this context.

Response – Done – (Figure 1, improved – Figure 2, we added SWOT analysis)

Minor points:

  • Subsections 1.1. and 1.2. are very short and both refer to AI. For this reason, I would suggest eliminating the division, combining both parts, and placing them in the Introduction where AI is first introduced (e.g. lines 56-57).

Response – Done (we placed them in the Introduction where AI is first introduced)

  • -Line 186: Perhaps this is "the area under the receiver operating characteristic (ROC) curve, that is AUC". Please check.

Response – Done (we change it to AUC)

Reviewer 4 Report

Comments and Suggestions for Authors

With the added quantative analysis, I believe this paper presents good summary on potential AI applciation for polypharmacy management in different subfileds. I don't have any further questions on it. 

Author Response

Thank you for your valuable feedback. We appreciate your comments in round one, which played an important role in improving the manuscript.